# HYBRID-REGRESSIVE NEURAL MACHINE TRANSLATION

## ABSTRACT

Although the non-autoregressive translation model based on iterative refinement has achieved comparable performance to the autoregressive counterparts with faster decoding, we empirically found that such aggressive iterations make the acceleration rely heavily on small batch size (e.g., 1) and computing device (e.g., GPU). By designing synthetic experiments, we highlight that iteration times can be significantly reduced when providing a good (partial) target context. Inspired by this, we propose a two-stage translation prototype – Hybrid-Regressive Translation (HRT). HRT first jumpily generates a discontinuous sequence by autoregression (e.g., make a prediction every $k$ tokens, $k > 1$). Then, with the help of the partially deterministic target context, HRT fills all the previously skipped tokens with one iteration in a non-autoregressive way. The experimental results on WMT'16 En↔Ro and WMT'14 En↔De show that our model outperforms the state-of-the-art non-autoregressive models with multiple iterations, and the original autoregressive models. Moreover, compared with autoregressive models, HRT can be steadily accelerated 1.5 times regardless of batch size and device.

## 1 INTRODUCTION

Although autoregressive translation (AT) has become the *de facto* standard for Neural Machine Translation (Bahdanau et al., 2015), its nature of generating target sentences sequentially (e.g., from left to right) makes it challenging to respond quickly in a production environment. One straightforward solution is the non-autoregressive translation (NAT) (Gu et al., 2017), which predicts the entire target sequence in one shot. However, such one-pass NAT models lack dependencies between target words and still struggles to produce smooth translations, despite many efforts developed (Ma et al., 2019; Guo et al., 2019a; Wang et al., 2019b; Shao et al., 2019; Sun et al., 2019).

Recent studies show that extending one-pass NAT to multi-pass NAT, so-called iterative refinement (IR-NAT), is expected to break the performance bottleneck (Lee et al., 2018; Ghazvininejad et al., 2019; Gu et al., 2019; Guo et al., 2020; Kasai et al., 2020a). Unlike one-pass NAT, which outputs the prediction immediately, IR-NAT takes the translation hypothesis from the previous iteration as a reference and regularly polishes the new translation until achieving the predefined iteration count $I$ or no changes appear in the translation. Compared with AT, IR-NAT with I=10 runs 2-5 times faster with a considerable translation accuracy, as reported by Guo et al. (2020).

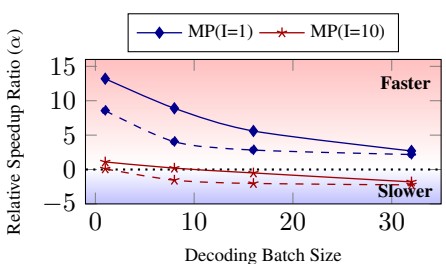

Figure 1: Relative speedup ratio ($\alpha$) compared MP with AT on GPU (solid) and CPU (dashed). The value of $\alpha$ denotes running faster (positive) or slower (negative) $|\alpha|$ times than AT.

However, we highlight that the fast decoding of IR-NAT heavily relies on *small batch size* and *GPU*, which is rarely mentioned in prior studies [1]. Without loss of generality, we take Mask-Predict (MP)

---

[1] Unfortunately, such a decoding setting is not common in practice. NMT systems deployed on GPUs tend to use larger batches to increase translation throughput, while the batch size of 1 is used more frequently in offline systems running on CPUs. e.g., smartphones.

(Ghazvininejad et al., 2019) as an example, a typical IR-NAT paradigm based on the conditional masked language model. Figure 1 illustrates that when the batch exceeds 8, MP(I=10) is already running slower than AT, and the situation is even worse on CPU. Further analysis shows that the increase in batch size leads to the efficiency degradation of parallel computing in NAT models [2].

To tackle this problem, we first design a synthetic experiment to understand the relationship between target context and iteration times. We mask some proportion tokens on the translation generated by a pretrained AT and take it as the decoder input of the pretrained MP. Then we surprisingly found that even masking 70% AT hypothesis, and the remaining target context can help MP(I=1) to compete with the standard MP(I=10) (Figure 2). This result confirms that decoding with multiple iterations in NAT is unnecessary when providing a good (partial) reference hypothesis.

Inspired by this, we propose a two-stage translation prototype——Hybrid-Regressive Translation (HRT). After encoding, HRT first uses an autoregressive decoder (called Skip-AT) to produce a discontinuous translation hypothesis. Concretely, at decoding step $i$, the SKip-AT decoder immediately predicts the $(i + k)$-th token $y_{i+k}$ without generating $y_{i+1}, \ldots, y_{i+k-1}$, where k is a hyperparameter and $k > 1$. Then, a non-autoregressive decoder like MP (called Skip-MP) predicts previously skipped tokens with one iteration according to the deterministic context provided by Skip-AT. Since both Skip-AT and Skip-MP share the same model parameters, HRT does not increase parameters significantly. To train HRT effectively and efficiently, we further propose *joint training guided by curriculum learning* and *mixed distillation*. Experimental results on WMT En↔Ro and En↔De show that HRT is far superior to existing IR-NATs and achieves comparable or even better accuracy than the original AT [3] with a consistent 50% decoding speedup on varying batch sizes and devices (GPU, CPU).

## 2 BACKGROUND

Given a source sentence $\boldsymbol{x} = \{x_1, x_2, \ldots, x_M\}$ and a target sentence $\boldsymbol{y} = \{y_1, y_2, \ldots, y_N\}$, there are several ways to model $P(\boldsymbol{y}|\boldsymbol{x})$:

**Autoregressive translation (AT)** is the dominant approach in NMT, which decomposes $P(\boldsymbol{y}|\boldsymbol{x})$ by chain rules:

$$P(\boldsymbol{y}|\boldsymbol{x}) = \prod_{t=1}^{N} P(y_t|\boldsymbol{x}, y_{<t}) \tag{1}$$

where $y_{<t}$ denotes the generated prefix translation before time step $t$. However, the existence of $y_{<t}$ requires the model must wait for $y_{t-1}$ to be produced before predicting $y_t$, which hinders the possibility of parallel computation along with time step.

**Non-autoregressive translation (NAT)** is first proposed by Gu et al. (2017), allowing the model to generate all target tokens simultaneously. NAT replaces $y_{<t}$ with target-independent input $\boldsymbol{z}$ and rewrites Eq. 1 as:

$$P(\boldsymbol{y}|\boldsymbol{x}) = P(N|x) \prod_{t=1}^{N} P(y_t|\boldsymbol{x}, \boldsymbol{z}) \tag{2}$$

In Gu et al. (2017), they monotonically copy the source embedding as $\boldsymbol{z}$ according to a fertility model. Subsequently, the researchers developed more advanced methods to enhance $\boldsymbol{z}$, such as adversarial source embedding (Guo et al., 2019a), reordered source sentence (Ran et al., 2019), latent variables (Ma et al., 2019; Shu et al., 2019) etc, but there still is a huge performance gap between AT and NAT.

**Iterative refinement based non-autoregressive translation (IR-NAT)** extends the traditional one-pass NAT by introducing the multi-pass decoding mechanism (Lee et al., 2018; Ghazvininejad et al., 2019; Gu et al., 2019; Guo et al., 2020; Kasai et al., 2020a). IR-NAT applies a conversion function

---

[2]Early experiment shows that when the batch size increases from 1 to 32, the latency of AT is reduced by 22 times, while MP(I=10) only reduces by four times. Latency is measured by the average time of translating a sentence on a constant test set. See Appendix A for details.

[3]Thanks to the proposed training algorithm, a single HRT model can support both hybrid-regressive decoding and autoregressive decoding at inference. Here, the AT model refers to the autoregressive teacher model that generates the distillation data.

$\mathcal{F}$ on the deterministic hypothesis of previous iteration $\boldsymbol{y}\prime$ as the alternative to $\boldsymbol{z}$. Common implementations of $\mathcal{F}$ include identity (Lee et al., 2018), random masking (Ghazvininejad et al., 2019) or random deletion (Gu et al., 2019) etc. Thus, we can predict $\boldsymbol{y}$ by:

$$P(\boldsymbol{y}|\boldsymbol{x}) = \prod_{t=1}^{N'} P(y'_{m(t)}|\boldsymbol{x}, \mathcal{F}(\boldsymbol{y}\prime)) \qquad (3)$$

where $N'$ is the number of refined tokens in $\mathcal{F}(\boldsymbol{y}\prime)$, m(t) is the real position of $t$-th refined token in $\boldsymbol{y}\prime$. In this way, the generation process of IR-NAT is simple: first, the NAT model produces an inaccurate translation as the initial hypothesis, and then iteratively refines it until converge or reaching the maximum number of iterations.

**Mask-Predict (MP)** is a typical instance of IR-NAT, trained by a conditional masked language model objective like BERT (Devlin et al., 2019). In this work, we use MP as the representation of IR-NAT due to its excellent performance and simplification. In MP, $\mathcal{F}$ randomly masks some tokens over the sequence in training but selects those predicted tokens with low confidences at inference.

## 3   IS ITERATIVE REFINEMENT ALL YOU NEED?

As mentioned earlier, IR-NAT with multiple iterations slows down severely in some cases. It is natural to think of reducing iterations to alleviate it. This section starts from synthetic experiments on WMT'16 En→Ro and WMT'14 En→De to verify the assumption that a sufficiently good decoder input can help reduce iterations. Here we construct the "good" decoder input from the translation hypothesis produced by an AT model.

**Models** We use the official MP models released by Ghazvininejad et al. (2019) [4]. Since the authors did not publish their AT baselines, we use the same data to retrain AT models with the standard Transformer-Base configuration (Vaswani et al., 2017) and obtain comparable performance with theirs (see Appendix B for more details).

**Decoding** AT models decode with beam sizes of 5 on both tasks. Then, we replace a certain percentage of AT translation tokens with *<mask>* and use it as input to the MP model (see below for replacement strategy). Unlike the standard MP model that uses a large beam size (e.g., 5) and iterates several times (e.g., 10), the MP model used here only iterates once with beam size 1. We substitute all input *<mask>* with MP's predictions to obtain the final translation. We report case-sensitive tokenized BLEU score by *multi-bleu.perl*.

**Mask Strategy** We tested 4 strategies to mask AT translations: `Head`, `Tail`, `Random` and `Chunk`. Given the masking rate $p_{mask}$ and the translation length $N$, the number of masked tokens is $N_{mask}=\max(1, \lfloor N \times p_{mask} \rfloor)$. Then `Head`/`Tail` always masks the first/last $N_{mask}$ tokens, while `Random` masks the translation randomly. `Chunk` is slightly different from the above strategies. It first divides the target sentence into $C$ chunks, where $C = \mathrm{Ceil}(N/k)$ and k is the chunk size. Then in each chunk, we retain the first token, but mask other k-1 tokens. Thus, the actual masking rate in `Chunk` is 1-1/k instead of $p_{mask}$. To exclude randomness, we ran `Random` three times with different seeds and report the average results.

### 3.1   RESULTS

The experimental results are illustrated in Figure 2, where we can see that:

**A balanced bidirectional context is critical.** Compared with `Tail` and `Head`, it is obvious that `Rand` and `Chunk` both have better performance. We attribute it to the benefit of the bidirectional context in `Rand` and `Chunk` (Devlin et al., 2019), because `Tail` and `Head` can only provide unidirectional context (i.e., prefix or suffix). In addition, compare `Chunk` with `Random`, we find that `Chunk` is moderately but consistently superior to `Random`, even if more tokens are masked. For instance, on the WMT En-De task, when the chunk size is 4 (the masking rate is 75%), the BLEU score of `Chunk` is 27.03, which is +0.3 BLEU higher than that of `Random` with the masking rate of

---

[4]`https://github.com/facebookresearch/Mask-Predict`

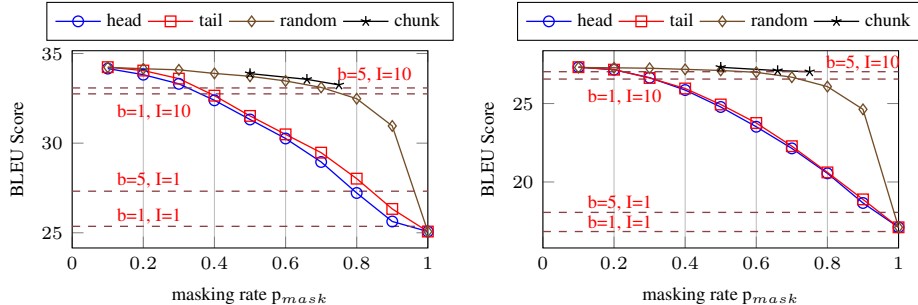

Figure 2: Comparison of four masking strategies {Head, Tail, Random, Chunk} in synthetic experiments on WMT En→Ro (Left) and En→De (Right) test sets. For Chunk, we test the chunk size from {2, 3, 4}. Dashed lines represent Mask-Predict's scores reported by Ghazvininejad et al. (2019). $b$ stands for "beam size" while $I$ stands for "the number of iterations".

Table 1: Examples of training samples. $<s2>$ is a special $$ for k=2. $<m>$ is short for $<mask>$.

| Mode | Input | Target |
|---|---|---|
| AT | $, y_1, y_2, y_3, y_4$ | $y_1, y_2, y_3, y_4, $ |
| MP | $y_1, <m>, <m>, y_4, <m>$ | PAD, $y_2, y_3$, PAD, $$ |
| Skip-AT (k=2) | $<s2>, y_2, y_4$ | $y_2, y_4, $ |
| Skip-MP (k=2) | $<m>, y_2, <m>, y_4, <m>, $ | $y_1$, PAD, $y_3$, PAD, $$, PAD |

70%. Because the difference between Chunk and Random lies only in the distribution of $<mask>$, this experiment indicates that making $<mask>$ uniformly on sequence is better than random [5].

**Small beams and one iteration are sufficient.** Compared with the standard MP with the beam size of 5 and 10 iterations, it is interesting to find that even if only 30%-40% of the AT translations are exposed, our MP using greedy search and one iteration can achieve quite comparable performance.

## 4 HYBRID-REGRESSIVE TRANSLATION

A limitation in the above synthetic experiment is that the MP decoder input comes from an AT hypothesis, which is impossible in practice [6]. To solve this problem as well as inspired by the Chunk strategy's success, we propose a two-stage translation paradigm called Hybrid-Regressive Translation (HRT). Briefly speaking, HRT can autoregressively generate a discontinuous sequence with chunk size $k$ (stage I), and non-autoregressively fill the skipped tokens (stage II) in one model. Thus, the standard AT can be regarded as the special case of HRT when k=1 without stage II.

### 4.1 ARCHITECTURE

**Overview.** Our HRT consists of three parts: encoder, Skip-AT decoder (for stage I), and Skip-MP decoder (for stage II). All components adopt the Transformer architecture (Vaswani et al., 2017): the encoder contains self-attention sublayers and feedforward sublayers, and additional cross-attention sublayers are added to the decoder. The two decoders have the same network structure and share model parameters, leading to the same parameter size compared to the standard MP. The only difference between the two decoders lies in the masking mode in the self-attention sublayer. The Skip-AT decoder masks future tokens to guarantee strict left-to-right generation, while the Skip-MP decoder eliminates this limitation to leverage the bi-directional context.

**Simplified relative position representation.** Another difference from the standard MP architecture is that our decoder self-attention equips with relative position representation (RPR) (Shaw et al.,

---

[5]Chunk guarantees that each unmasked token (except the first or last one in the sequence) can meet two deterministic tokens within the window size of $k$. However, in extreme cases, when all $<mask>$ happen to concentrate on the left/right side of the sequence, Random will degrade into Head/Tail

[6]We can directly return AT predictions as translation results without going through MP.

2018) to enable the model to capture the positional relationship between words easily [7]. Precisely, the decoder self-attention with RPR is calculated by:

$$O_i = \text{Softmax}\Big(\frac{Q_i(K^T + R_i)}{\sqrt{d_k}}\Big)V \tag{4}$$

where $R_i$ is the relative position embedding [8]. Note that Eq. 4 only injects the relative positional representation in Key ($K^T + R_i$) without involving Value $V$. We found that this simplification has no negative impact on performance but significantly saves memory footprint.

**No target length predictor.** Most previous NAT methods need to jointly train the translation model with an independent translation length predictor. However, such a length predictor is unnecessary for us because the translation length is a by-product of Skip-AT, e.g., $N_{nat}$=k$\times N_{at}$, where $N_{at}$ is the sequence length produced by Skip-AT [9]. Another bonus is that we can avoid carefully tuning the weighting coefficient between the length loss and the token prediction loss.

## 4.2 TRAINING

Training HRT models is not trivial because a single HRT model needs to learn to generate sequences by both autoregression and non-autoregression. This section will introduce three details of training HRT models, including chunk-aware training samples, curriculum learning, and mixed distillation. We describe the entire training algorithm in Appendix C.

**Chunk-aware training samples.** As listed in Table 1, the training samples for Skip-AT and Skip-MP are different from the standard AT and MP. Compared with AT, Skip-AT shrinks the sequence length from N to N/k. It should be noted that, although the sequence feeding to Skip-AT is shortened, the input position still follows the original sequence rather than the surface position. For example, in Table 1, the position of Skip-AT input ($<s2>, y_2, y_4$) is (0, 2, 4), instead of (0, 1, 2). Moreover, MP has the opportunity to mask any token over the target sequence without considering the position. However, the masking pattern in Skip-MP is deterministic, i.e., masking all non-first tokens in each chunk. Therefore, we can say that the training sample for Skip-AT and Skip-MP is in a chunk-aware manner.

**Curriculum learning.** Unfortunately, direct joint training of Skip-AT and Skip-MP is problematic because the chunk-aware training samples cannot make full use of all the tokens in the sequence. For example, in Table 1, the target tokens $y_1$ and $y_3$ have no chance to be learned as the decoder input of either Skip-AT or Skip-MP. However, there is no such problem in AT and MP. Therefore, we propose to gradually transition from joint training {AT, MP} to {Skip-AT, Skip-MP} through curriculum learning (Bengio et al., 2009). In other words, the model is trained from chunk size 1 to chunk size k (k>1). More concretely, given a batch of original sentence pairs $B = (\mathbf{x}_i, \mathbf{y}_i)|_{i=1}^n$ and let the proportion of chunk size k in $B$ be $p_k$, we start with $p_k = 0$ and construct the training samples of AT and MP for all pairs. And then we gradually increase $p_k$ to introduce more learning signals for Skip-AT and Skip-MP until $p_k$=1. In implement, we schedule $p_k$ by:

$$p_k = \Big(\frac{t}{T}\Big)^\lambda \tag{5}$$

where $t$ and $T$ are the current and total training steps, respectively. $\lambda$ is a hyperparameter and we use $\lambda$=1 to increase $p_k$ in a linear manner for all experiments.

**Mixed Distillation.** NAT models generally use the distillation data generated by AT models due to more smoothing data distribution (Zhou et al., 2020). However, making full use of distillation data may miss the diversity in raw data. To combine the best of both worlds, we propose a simple and effective approach – *Mixed Distillation* (MixDistill). During training, MixDistill randomly samples a target sentence from the raw version $\boldsymbol{y}$ with probability $p_{raw}$ or its distillation version $\boldsymbol{y}^*$ with probability 1-$p_{raw}$, where $p_{raw}$ is a hyperparameter [10]. By learning from raw target sentences, we empirically found that HRT is less prone to over-fit in some simple tasks (e.g., WMT'16 En→Ro).

---

[7] We keep the sinusoidal absolute position embedding unchanged.

[8] $R_i = \text{Embed}(\text{clip}(i, 1), \ldots, \text{clip}(i, N))$, where $\text{clip}(i, j) = \max(w, \min(w, j-i))$, $w$ is the window size.

[9] More precisely, $N_{nat}$ here is the maximum target length rather than the realistic length because multiple $$ may be predicted in the last $k$ tokens.

[10] Training with full raw data or full distillation data can be regarded as the special case of MixDistill when $p_{raw}$=1 or $p_{raw}$=0.

Table 2: The BLEU scores of our proposed HRT and the baseline methods on four tasks. Unless otherwise stated, the used beam size is 5. "?" denotes dynamic iterations. "20L" stands for using a 20-layer encoder. "NPD" is short for Noisy Parallel Decoding. All HRT models only iterate once by non-autoregression.

| | System | Iterations | WMT'16 | | WMT'14 | |
|---|---|---|---|---|---|---|
| | | | En-Ro | Ro-En | En-De | De-En |
| ATs | Transformer | - | 34.25 | 34.40 | 27.45 | 31.86 |
| | Transformer-20L | - | - | - | 28.79 | 33.02 |
| | *Iterative* | | | | | |
| Existing NATs | Iterative Refinement (Lee et al., 2018) | ? | 29.66 | 30.30 | 21.54 | 25.43 |
| | Mask-Predict (Ghazvininejad et al., 2019) | 1 | 27.32 | 28.20 | 18.05 | 21.83 |
| | Mask-Predict (Ghazvininejad et al., 2019) | 10 | 33.08 | 33.31 | 27.03 | 30.53 |
| | LevTransformer (Gu et al., 2019) | ? | - | - | 27.27 | - |
| | JM-NAT (Guo et al., 2020) | 10 | 33.52 | 33.72 | 27.69 | 32.24 |
| | *Non-Iterative* | | | | | |
| | SAT (k=2) (Wang et al., 2018a) | - | - | - | 26.90 | - |
| | FCL-NAT (NPD 9) (Guo et al., 2019b) | 1 | - | - | 25.75 | 29.50 |
| Our models | HRT ($b_{at}$=1, $b_{mp}$=1) | 1 | 34.11 | 34.28 | 27.85 | 31.80 |
| | HRT ($b_{at}$=5, $b_{mp}$=1) | 1 | 34.36 | 34.55 | 27.98 | 31.93 |
| | HRT ($b_{at}$=5, $b_{mp}$=5) | 1 | **34.53** | **34.80** | 28.10 | 32.07 |
| | HRT-20L ($b_{at}$=1, $b_{mp}$=1) | 1 | - | - | 28.79 | 32.86 |
| | HRT-20L ($b_{at}$=5, $b_{mp}$=1) | 1 | - | - | 28.90 | 33.06 |
| | HRT-20L ($b_{at}$=5, $b_{mp}$=5) | 1 | - | - | **28.99** | **33.08** |

## 4.3 INFERENCE

After encoding, the Skip-AT decoder starts from $<sk>$ to autoregressively generate a discontinuous target sequence $\boldsymbol{y}_{at} = (z_1, z_2, \ldots, z_m)$ with chunk size $k$ until meeting $$. Then we construct the input of Skip-MP decoder $\boldsymbol{x}_{mp}$ by appending $k-1$ $<mask>$ before every $z_i$. The final translation is generated by replacing all $<mask>$ with the predicted tokens by Skip-MP decoder with one iteration. If there are multiple $$ existing, we truncate to the first $$. Note that the beam size $b_{at}$ in Skip-AT can be different from the beam size $b_{mp}$ in Skip-MP as long as st. $b_{at} \geq b_{mp}$. If $b_{at} > b_{mp}$, then we only feed the Skip-MP with the top $b_{mp}$ Skip-AT hypothesis. Finally, we choose the hypothesis with the highest score:

$$score(\hat{\boldsymbol{y}}) = \underbrace{\sum_{i=1}^{m}(z_i|\boldsymbol{x}, z_{<i})}_{\text{Skip-AT score}} + \underbrace{\sum_{i=0}^{m-1}\sum_{j=1}^{k-1}(\hat{y}_{i\times k+j}|\boldsymbol{x}, \boldsymbol{x}_{mp})}_{\text{Skip-MP score}} \quad (6)$$

where $z_i = \hat{y}_{i\times k}$. In Appendix D, we summarized the comparison with the existing three methods from the aspects of decoding step and calculation cost, including AT, MP, and semi-autoregressive translation (SAT) (Wang et al., 2018a). Besides, thanks to the joint training of chunk size 1 and $k$ simultaneously, the HRT model can also behave like a standard AT model by forcing decoding by chunk size one (denoted as $C_d$=1). In this way, we can only use the Skip-AT decoder to generate the entire sequence without the help of Skip-MP. Thus, $C_d$=1 can be regarded as the performance upper bound when the decoding chunk size is $k$ (denoted as $C_d$=$k$).

## 5 EXPERIMENTS

**Datasets.** We conducted experiments on four widely used tasks: WMT'16 English↔Romanian (En↔Ro, 610k) and WMT'14 English↔German (En↔De, 4.5M). We replicated the same data processing as Ghazvininejad et al. (2019) for fair comparisons.

**AT teachers for distillation.** Since Ghazvininejad et al. (2019) only release the distillation data of En↔Ro, not En↔De, we retrained the AT teacher models of En↔De to produce the distillation data. Specifically, Ghazvininejad et al. (2019) use Transformer-Large as the teacher, but we use the deep PreNorm Transformer-Base with a 20-layer encoder, which is faster to train and infer with comparable performance (Wang et al., 2019a).

**Models and hyperparameters.** We ran all experiments on 8 TITAN X (Pascal) GPUs. Unless noted otherwise, we use the chunk size k=2 and $\lambda$=1. $p_{raw}$=0.5 for En↔Ro and $p_{raw}$=0.8 for

En↔Ro according to validation sets. The windows size of RPR is 16 [11]. HRT models are finetuned on pretrained AT models for the same training steps [12]. Other training hyperparameters are the same as Vaswani et al. (2017) or Wang et al. (2019a) (using deep-encoder). Please refer to them for more details.

**Translation quality.** Table 2 reports the BLEU scores on four tasks. First of all, we can see that IR-NAT models significantly outperform those one-pass NAT models (e.g., SAT, FCL-NAT, FlowSeq). However, our small beam model ($b_{at}$=$b_{mp}$=1) can defeat the existing multiple-iteration models. Furthermore, when the beam sizes increase to 5, HRT equipped with a standard 6-layer encoder achieves +1.0 BLEU point improvement in En↔Ro compared to the previous best results (Guo et al., 2020). Even on harder En↔De tasks, we also outperform them with a 0.4 BLEU score. We can easily trade-off between performance and speed by using $b_{at}$=5 but $b_{mp}$=1. Not only NAT models, but also we are surprised to find that HRT can even surpass the AT models trained from scratch. We attribute it to two reasons: (1) HRT is fine-tuned on a well-trained AT model, making training easier; (2) Mixing up AT and NAT has a better regularization effect than training alone. Besides, in line with Guo et al. (2020), which demonstrate that the encoder is critical for NAT, we can obtain a further improvement of about +0.8 BLEU when using a deeper encoder.

**On distant language pair.** To verify our method can apply to distance language pairs, we conducted a new experiment on the Chinese-English (Zh-En) translation task. We use the same datasets as Wang et al. (2018b): the training data is selected from LDC Corpus and contains 1.8M sentence pairs; NIPS MT06 as the validation set and MT04, MT05, and MT08 as test sets. We train all models for 50k steps, and other training setting is the same as WMT En-De. The experimental results are listed in Table 3. We can see that HRT is superior to the original AT model and MP model again, which indicates that HRT is effective in both close and distant language pairs.

Figure 3: BLEU scores on the Chinese-English task.

| Model | MT04 | MT05 | MT08 |
|---|---|---|---|
| AT | 43.86 | 52.91 | 33.94 |
| MP(I=10) | 42.47 | 52.16 | 33.09 |
| HRT1-1 | 43.96 | 53.16 | 33.99 |
| HRT5-1 | 44.28 | 53.44 | 34.63 |
| HRT5-5 | **44.31** | **53.77** | **34.74** |

**Translation speed.** Unlike the previous works that only run the model on GPU with batch size of 1, we systematically test the decoding speed using varying batch sizes and devices on WMT'14 En→De test set (see Figure 4). By default, the beam size is 5. It can be seen that although HRT is slower than MP10 when running on GPU with a batch size of 1, MP10 dramatically slows down as the batch size increases. In contrast, HRT5-1 is consistently more than 50% faster than AT without changing with the environment. These results show that our HRT can be an effective and efficient substitute for AT and IR-NAT.

# 6 ANALYSIS

**Effect of Mixed Distillation.** In Table 3, we compared different data strategies, including raw data (Raw), sequence-level knowledge distillation (Dist.), and mixed distillation (Mix Dist.). Overall, Mix Dist. is superior to other methods across the board, which indicates that training with raw data and distillation data is complementary. In addition, we also find that the performance of the distillation data is lower than the raw data on En→Ro task, which is against the previous results. As interpreted by Zhou et al. (2020), we suspect that when the translation model is strong enough, training by distillation data completely may make the learning too easy and lead to over-fitting.

**Effect of chunk size.** We tested the chunk size k from {2, 3, 4}, and the results are listed in Table 4. Obviously, we can see that: (1) A large $k$ has more significant acceleration on GPU because fewer autoregressive steps are required; (2) As k increases, the performance of hybrid-regressive decoding

---

[11]For autoregressive baselines, adding RPR in the Transformer decoder did not bring obvious improvement over the vanilla Transformer. For example, on WMT'14 En→De, Transformer=27.45 and Transformer+RPR=27.34.

[12]Since HRT needs to train Skip-AT and Skip-MP jointly (please see Algorithm 1 in Appendix C), the wall-clock time is about two times longer than AT in the same training epochs. One more thing to note is that the officially released MP models are trained for 300k steps from scratch.

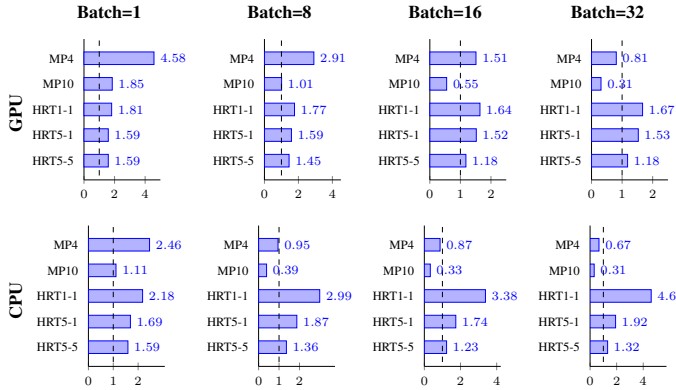

Figure 4: Comparison of decoding speed w.r.t. batch size and computing device on WMT'14 En→De task. The x-axis is the relative speed compared to the corresponding autoregressive models (dashed lines at x=1). GPU:TITAN X (Pascal), CPU:Intel(R) Xeon(R) E5-2680 v3 @ 2.50GHz. MP{#} denotes the MP model with # iterations. HRT{#1}−{#2} denotes HRT with $b_{at}$=#1 and $b_{mp}$=#2. Note that the BLEU score of MP4 is 25.94, which is significantly lower than that of HRT (e.g., the BLEU score of HRT1-1 is 27.85).

Table 3: Performance against different data strategies. $C_d$=1 represents decoding the HRT model in an autoregressive manner.

| Lang. | $C_d$ | Raw | Dist. | Mix Dist. |
|---|---|---|---|---|
| En→Ro | k | 33.92 | 33.41 | **34.53** |
|  | 1 | **34.29** | 33.41 | 34.27 |
| En→De | k | 26.37 | 28.00 | **28.10** |
|  | 1 | 27.60 | 28.42 | **28.51** |

Table 4: The effect of training by different chunk sizes. Latency is tested in batch size of 16 using $C_d$=k and $b_{at}$=$b_{mp}$=1.

| Chunk | BLEU | | Latency (sec.) | |
|---|---|---|---|---|
| (k) | $C_d= k$ | $C_d= 1$ | GPU | CPU |
| 2 | **34.11** | 33.86 | 20.0 | 70.5 |
| 3 | 31.15 | 33.78 | 13.0 | 54.6 |
| 4 | 28.22 | **34.12** | **12.2** | **53.9** |

drops sharply (e.g., k=4 is 6 BLEU points lower than k=2.), but $k$ has little effect on autoregressive modes. It indicates that the training difficulty of Skip-AT increases as $k$ gets bigger. We think that skip-generation may require more fancy model architecture or training method, which is left for our future work.

**Effect of decoding mode.** We tested the well-trained HRT model with two decoding modes: autoregressive ($C_d$=1) and hybrid-regressive ($C_d$=k). Concretely, We divided the test set of WMT'14 En→De into several groups according to the source sentence's length and then compared the two decoding modes in terms of translation speed and accuracy in each group (see Figure 5). First of all, we can see that regardless of the source sentence length, the running speed of $C_d$=k is consistently faster than $C_d$=1 on both GPU and CPU, thanks to the shorter autoregressive length. This advantage is more evident on CPU: When the source length is less than 10, $C_d$=k runs 1.6 times faster than $C_d$=1, while the speedup ratio increases to 2.0 when the source length > 50. As for accuracy, $C_d$=k has closed performance to $C_d$=1 when the length is between 10 and 30, but shorter or longer sentences will hurt the performance. This result indicates that if we dynamically adjust the decoding chunk size $C_d$ according to the source sentence's length, the HRT model can be expected to improve the performance further at the expense of a certain speed.

**Ablation study.** We also did an ablation study on WMT'16 En→Ro test set. As shown in Table 5, we can see that all introduced techniques help to improve performance. In particular, using mixed distillation prevents the HRT model from over-fitting and leads to +1.1 BLEU points improvement compared to the standard distillation (-MixDistill). In addition, the other three methods, including training the HRT model from a pretrained AT model (FT), using a relative positional representation on decoder (RPR), and using curriculum learning (CL), can bring about 0.3-0.4 BLEU improvements each. It should be noted that removing curriculum learning makes the trained HRT model fail to decode by $C_d$=1, whose BLEU score is only 5.18. Since the BLEU score decreases slightly (0.3-0.4 except -MixDistill) when each component is excluded independently, it is difficult to say

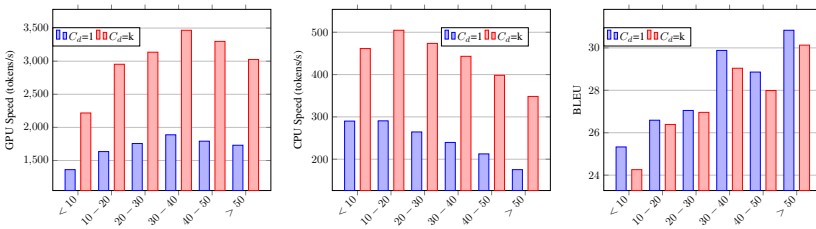

Figure 5: Translation speed (Left: GPU, Middle: CPU) and BLEU score (Right) against source sentence's length for different decoding modes in HRT (k=2): $C_d$=1 denotes decoding by autoregression, while $C_d$=k denotes hybrid-regressive decoding. Speed is measured at batch size=32, $b_{at}$=5, $b_{mp}$=1.

that the difference of BLEU is not caused by random fluctuation. To verify it, we try to exclude them all from the standard HRT (-ALL). Interestingly, the obtained model drops by 1.96 BLEU points, which is very close to the cumulative BLEU loss (2.13) of excluding each component separately. It indicates that these newly introduced components are complementary. In addition, we also test these methods in MP training. Please see Appendix E for details.

Table 5: Ablation study on WMT'16 En→Ro test set.

| System | AT | HRT | −FT | −RPR | −MixDistill | −CL($q_k$=1.0) | −ALL |
|---|---|---|---|---|---|---|---|
| **BLEU** | 34.25 | 34.53 | 34.14 | 34.22 | 33.41 | 34.22 | 32.57 |

# 7 RELATED WORK

**Iterative refinement.** Lee et al. (2018) first extend NAT from the conventional one-pass manner to the multi-pass manner. They add an additional decoder to learn to recover from a collapsed target sequence to gold one. Mask-Predict (Ghazvininejad et al., 2019) simplifies the two-decoder structure by introducing the conditional masked language model objective. During each iteration, Mask-Predict retains partial inputted target tokens according to the prediction confidence, while LevTransformer (Gu et al., 2019) uses multiple discriminators to determine the edited tokens.

**Combination of AT and NAT.** The idea of incorporating AT in the NAT model is not new (Kaiser et al., 2018; Ran et al., 2019; Akoury et al., 2019). The main difference from existing methods lies in the content of AT output, such as latent variables (Kaiser et al., 2018), reordered source tokens (Ran et al., 2019), syntactic labels (Akoury et al., 2019) etc. In contrast, our approach uses the deterministic target tokens, which has been proven effective in Ghazvininejad et al. (2019).

**Decoding acceleration.** In addition to transforming the decoding paradigm from autoregressive to non-autoregressive, there are many works to explore how to achieve faster decoding from other aspects. Zhang et al. (2018c;b) propose to optimize the beam search progress by recombining or pruning the translation hypothesis. Considering the network architecture, Zhang et al. (2018a) use light AAN instead of the standard self-attention module; Xiao et al. (2019) share the self-attention weight matrix across decoder layers; Kasai et al. (2020b) suggest using deep-encoder and shallow-decoder network to keep high BLEU score and low delay. Moreover, some common model compression techniques, such as distillation (Kim & Rush, 2016) and quantization (Bhandare et al., 2019; Lin et al., 2020), have also helpful for acceleration. However, the above methods mainly focus on the traditional autoregressive translation, which is orthogonal to our work.

# 8 CONCLUSION

We have pointed out that NAT, especially IR-NAT, cannot efficiently accelerate decoding when using a large batch or running on CPUs. Through a well-designed synthetic experiment, we highlighted that given a good decoder input, the number of iterations in IR-NAT could be dramatically reduced. Inspired by this, we proposed a two-stage translation paradigm HRT to combine AT and NAT's advantages. The experimental results show that HRT owning equivalent or even higher accuracy and 50% acceleration ratio on varying batches and computing devices is a good substitute for AT.

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

## A  SPEED DEGRADATION ANALYSIS OF NAT MODEL

Table 6: Time-consuming of decoding *newstest2014* by different batch sizes on Titian X GPU.

| Model | BH=1 | BH=8 | BH=16 | BH=32 |
|:---:|:---:|:---:|:---:|:---:|
| AT | 962s | 151s | 78s | 43s |
| MP ($I$=10) | 464s | 125s | 119s | 129s |

For the NAT model, we assume that the computation cost of each iteration is proportional to the size of decoder input tensor $(BH \times BM, L, H)$, where $BH$ is the batch size, $BM$ is the beam size, $L$ is the predicted target length, and $H$ is the network dimension. In this way, the total cost of $I$ iterations (generally, $I < L$) is $C_{nat} \propto I \times \mathcal{O}(BH \times BM \times L \times H)$. For convenience, we omit $BM$ and $H$ and simplify $C_{nat}$ to $I \times \mathcal{O}(BH \times L)$. Likely, the computational cost of AT model is about $C_{at} \propto L \times \mathcal{O}(BH \times 1)$ [13]. Then, we can denote the speedup ratio $r$ as $r = \frac{C_{at}}{C_{nat}} = \frac{L}{I} \times \frac{\mathcal{O}(BH \times 1)}{\mathcal{O}(BH \times L)}$. Thus, fewer iterations (small $I$) and faster parallel computation (large $\frac{\mathcal{O}(BH \times 1)}{\mathcal{O}(BH \times L)}$) are the keys to IR-NAT.

However, in practice, we find it difficult to increase $\frac{\mathcal{O}(BH \times 1)}{\mathcal{O}(BH \times L)}$, especially in larger batches. As shown in Table 6, the direct evidence is that when decoding the test set of the WMT'14 En→De task, the time spent by AT decreases with the increase of batch size, while MP($I$=10) cannot benefit from this. We test it on Titian X GPU and report the average of 3 runs. $BM$=5. Note that CPU has similar results. Specifically, we can see that when $BH$ increases from 1 to 32, AT's latency reduces by 962/43 (about 22) times, while MP ($I$=10) only reduces by 464/129 (about 4) times. It means that $\frac{\mathcal{O}(BH \times 1)}{\mathcal{O}(BH \times L)}$ becomes lower as $BH$ increases. Until $\frac{\mathcal{O}(BH \times 1)}{\mathcal{O}(BH \times L)} < \frac{I}{L}$, NAT will start to be slower than AT.

## B  AT TRANSFORMER IN SYNTHETIC EXPERIMENTS

Table 7: Performance of autoregressive models in the synthetic experiment.

| AT Transformer | En-Ro | En-De |
|:---:|:---:|:---:|
| Vaswani et al. (2017) | - | 27.3 |
| Ghazvininejad et al. (2019) | 34.28 | 27.74 |
| Our implementation | 34.25 | 27.45 |

In the synthetic experiment, we trained all AT models with the standard Transformer-Base configuration: layer=6, dim=512, ffn=2048, head=8. The difference from Ghazvininejad et al. (2019) is that they trained the AT models for 300k steps, but we updated 50k/100k steps on En→Ro and En→De, respectively. Although fewer updates, as shown in Table 7, our AT models have comparable performance with theirs.

## C  TRAINING ALGORITHM

Algorithm 1 describes the process of training the HRT model. The HRT model is pre-initialized by a pre-trained AT model (Line 1). During training, the training batch $\boldsymbol{B}_i$ randomly select a raw target sentence $\boldsymbol{Y}_i$ or its distilled version $\boldsymbol{Y'}$ (Line 4-6). Then according to Eq. 5, we can divide $\boldsymbol{B}$ into two parts: $\boldsymbol{B}_{c=1}$ and $\boldsymbol{B}_{c=k}$, where $|\boldsymbol{B}_{c=k}|/|\boldsymbol{B}| = p_k$ (Line 7-8). Next, we construct four kinds of training samples based on corresponding batches: $\boldsymbol{B}_{c=k}^{at}$, $\boldsymbol{B}_{c=1}^{at}$, $\boldsymbol{B}_{c=k}^{mp}$ and $\boldsymbol{B}_{c=1}^{mp}$. Finally, we collect all training samples together and accumulate their gradients to update the model parameters, which results in the batch size being twice that of standard training.

---

**Algorithm 1** Training Algorithm for Hybrid-Regressive Translation

---

**Input:** Training data $D$ including distillation targets, pretrained AT model $\mathrm{M}_{at}$, chunk size $k$, mixed distillation rate $p_{raw}$

**Output:** Hybrid-Regressive Translation model $\mathrm{M}_{hrt}$

1: $\mathrm{M}_{hrt} \leftarrow \mathrm{M}_{at}$              ▷ finetune on pre-trained AT
2: **for** $t$ in $1, 2, \ldots, T$ **do**
3:   $\boldsymbol{X} = \{\boldsymbol{x}_1, \ldots, \boldsymbol{x}_n\}, \boldsymbol{Y} = \{\boldsymbol{y}_1, \ldots, \boldsymbol{y}_n\}, \boldsymbol{Y}' = \{\boldsymbol{y}'_1, \ldots, \boldsymbol{y}'_n\} \leftarrow$ fetch a batch from $D$
4:   **for** $i$ in $1, 2, \ldots, n$ **do**
5:    $\boldsymbol{B}_i = (\boldsymbol{X}_i, \boldsymbol{Y}_i^*) \leftarrow$ sampling $\boldsymbol{Y}_i^* \sim \{\boldsymbol{Y}_i, \boldsymbol{Y}_i'\}$ with $P(\boldsymbol{Y}_i) = p_{raw}$   ▷ mixed distillation
6:   **end for**
7:   $p_k \leftarrow$ get the chunk-aware proportion by Eq. 5       ▷ curriculum learning
8:   $\boldsymbol{B}_{c=k}, \boldsymbol{B}_{c=1} \leftarrow \boldsymbol{B}_{:\lfloor n \times p_k \rfloor}, \boldsymbol{B}_{\lfloor n \times p_k \rfloor:}$         ▷ split batch
9:   $\boldsymbol{B}_{c=k}^{at}, \boldsymbol{B}_{c=k}^{mp} \leftarrow$ construct {Skip-AT, Skip-MP} training samples based on $\boldsymbol{B}_{c=k}$
10:   $\boldsymbol{B}_{c=1}^{at}, \boldsymbol{B}_{c=1}^{mp} \leftarrow$ construct {AT, MP} training samples based on $\boldsymbol{B}_{c=1}$
11:   Optimize $\mathrm{M}_{hrt}$ using $\boldsymbol{B}_{c=k}^{at} \cup \boldsymbol{B}_{c=1}^{at} \cup \boldsymbol{B}_{c=k}^{mp} \cup \boldsymbol{B}_{c=1}^{mp}$     ▷ joint training
12: **end for**

---

Table 8: Compare hybrid-regressive translation (HRT) to autoregressive translation (AT), iterative refinement based non-autoregressive translation (IR-NAT), and semi-autoregressive translation (SAT). $Q(i)$ denotes the computation cost in autoregressive mode when producing the $i$-th token (e.g., the prefix length is $i-1$). $\hat{Q}_b(i)$ denotes the computation cost in non-autoregressive mode when producing $i$ tokens by one shot with a beam size of $b$. $I = 4 \sim 10$, $k$ is generally 2.

| Method | Steps | Computing Cost |
|--------|-------|----------------|
| AT | L | $\sum_{i=1}^{L} Q(i)$ |
| IR-NAT | I | $I \times \hat{Q}_{b=5}(L)$ |
| SAT | L/k | $L/k \times (\hat{Q}_{b=5}(k) + \epsilon)$ |
| HRT | L/k + 1 | $\sum_{i=1}^{L/k} Q(i \times k) + \hat{Q}_{b=1}(L)$ |

## D   COMPUTATION COMPLEXITY

In Table 8, we summarized the comparison with autoregressive translation (AT), iterative refinement based non-autoregressive translation (IR-NAT) and semi-autoregressive translation (SAT) (Wang et al., 2018a).

**AT.** Although both HRT and AT contain a slow autoregressive generation process, HRT's length is k times shorter than AT. Considering that the computational complexity of self-attention is quadratic with its length, HRT can save more time in autoregressive decoding.

**IR-NAT.** Since Skip-AT provides a high-quality target context, HRT does not need to use large beam size and multiple iterations like IR-NAT. The experimental results also show that our light NAT can make up for the increased cost in Skip-AT and can achieve stable acceleration regardless of the decoding batch size and running device.

**SAT.** SAT generates segments locally by non-autoregression, but it is still autoregressive between segments. We claim that the SAT reduces the decoding steps by $k$, but each token's calculation remains unchanged. In other words, in the time step $i$, there are $i-1$ tokens used for self-attention. By contrast, only $i/k$ tokens are involved in our Skip-AT.

## E   APPLY THE OPTIMIZATION METHODS TO MASK-PREDICT

We conducted experiments on the WMT En-De task to verify whether the optimization methods used in HRT training are complementary to MP, including fine-tuning from the pre-trained autoregressive

---

[13]While the decoder self-attention module considers the previous $i$ tokens, we omit it here for the sake of clarity.

Table 9: Apply the optimization methods used in HRT training to MP. BLEU scores are evaluated on the WMT'14 En→De test set.

| Method | Steps | BLEU |
|---|---|---|
| MP (official) | 300k | 27.03 |
| HRT ($b_{at}$=5, $b_{mp}$=1) | 100k | 27.98 |
| MP + FT + RPR + MD | 100k | 27.02 |
| MP + FT + RPR + MD | 160k | 27.32 |
| MP + FT + RPR + MD | 300k | 27.63 |

model (FT), relative positional representation (RPR), and mixed distillation (MD). The joint training of Skip-AT and Skip-MP through curriculum learning is not involved, because it is incompatible with MP training. Table 9 shows that the optimization methods used in HRT training are complementary to MP training. With the help of FT+RPR+MD, our MP model with 100k steps can achieve almost the same BLEU score as the officially released model with 300k steps. What's more, when we train more steps, our MP is improved by +0.61 BLEU points compared with the official model, but still falls behind our HRT model.

