# OpenReview forum: "Hybrid-Regressive Neural Machine Translation"
_ICLR.cc/2021/Conference — Reject_

### Official Review · AnonReviewer3 · 2020-10-27
**the studied problem is novel, but the comparison is not fully convincing.**

**Rating:** 5
**Confidence:** 4

**Review:**

Non-autoregressive decoder (NAT) greatly improves translation efficiency, but often relies on iterative refinement (IR) to retain the translation performance. Unfortunately, this strategy makes the decoding efficiency of IR-based NAT much more sensitive to batch size and computing device: with larger batch size and CPU, IR-based NAT runs even slower than the standard auto-regressive decoder (AT).

This paper targets at alleviating this IR bottleneck by proposing hybrid-regressive translation (HRT), which combines AT and NAT in two stages. AT in the first stage aims at offering an initial coarse target hypothesis, implemented by predicting target words at every k position, to serve as a better target context for NAT such that NAT can produce high-quality translation with only one iteration in the second stage. The authors introduce several tricks to optimize HRT, including joint training, mixed distillation and curriculum learning etc. Experiments on two WMT translation tasks (four translation directions) demonstrate the effectiveness of HRT, which consistently accelerates decoding by >50% compared to the AT baseline and yields comparable or even better translation quality.

Overall, this IR issue is under-studied in the literature, which deserves more attention. But I am not fully convinced by the current experiments, especially the comparisons.

My major concerns:

= Firstly, only comparing with MP10 is not enough. HRT consists of two parts. The first part AT acts very similar to the semi-autoregressive model (Wang et al, EMNLP 2018), which should be treated as a baseline. The second part, NAT or MP used in experiments, involves different variants when different iterations are used. For example, Figure 2 in Mask-Predict paper (Ghazvininejad et al, EMNLP 2019) discovers the trade-off between speedup and translation quality. Could you please show the Pareto frontier to prove that HRT reaches a better trade-off compared to MP? Besides, one thing should also be noticed: the speedup yielded by HRT seems less charming compared to recent NAT models, such as Levenshetain Transformer 3x-4x (Gu et al, NeurIPS 2019), and JM-NAT (k = 10) 5.73x (Guo et al, ACL 2020) where both models achieve similar translation quality to the AR baseline.

= If I understand correctly, the advantage of HRT in handling different batch sizes and computing devices mainly comes from its usage of one-pass NAT. In Figure 3, we observe that HRT1-1 produces clearly larger speedups as batch size increases from the CPU graphs. However, we observe a reversed trend with MP1 in Figure 1. Could you give readers some explanation?

= Although this paper mainly focuses on improving NAT, efforts on accelerating decoding is not limited to NAT. Relevant studies, which should be at least discussed in related work, include model quantization [1] and simplified decoder [2, 3], to name a few, are all missed in the paper. In particular, some models like AAN [2] and Deep encoder+Shallow decoder[3] can already produce similar speedups (might be smaller) with comparable translation performance. These models do not rely on knowledge distillation and those complex optimization tricks used for HRT. Could you please explain the advantage of HRT over these models?

= The claim “Chunk is superior to Rand, which indicates that the balanced distribution of deterministic tokens is necessary” (above Section 4) is strong. From Figure 2, what I see is that Chunk performs slightly better than Random. It’s hard to conclude that the balanced distribution is *necessary*.

My minor concerns:

= How did you handle the positional encoding in AT of HRT? Did you consider the interval k?

= The proposed optimization tricks are somehow orthogonal to HRT itself. What if you apply them to the baselines, like MP and semi-autoregressive model?

= How many runs did you perform when reporting the speed numbers in Figure 3?

= I cann’t find the number *34.08* (Table 3) in Table 2 for En->Ro translation. Is there something wrong?

= The citation format should be adjusted accordingly. The authors always use the format of “author (year)”.

= It would be better to see more results on the translation of distant language pairs, such as WMT English-Chinese.

[1] Bhandare et al.  Efficient 8-bit quantization of transformer neural machine language translation model. 2019

[2] Zhang et al. Accelerating neural transformer via an average attention network. ACL 2018

[3] Kasai et al. Deep Encoder, Shallow Decoder: Reevaluating the Speed-Quality Tradeoff in Machine Translation. 2020

---

> ### Author Response · Authors · 2020-11-16
> **Response to Reviewer #3**
>
> Thanks for your insightful reviews and we appreciate your valuable suggestions! We address your concerns and questions as follows:
>
> ### 1. Semi-autoregressive baseline  ###
>
> We thank the reviewer for pointing out this issue. We have compared the semi-autoregressive model (SAT) with us in Table 2 in terms of the BLEU score. We didn't compare with SAT in detail because  In addition, although we didn't reimplement SAT as a baseline, we analysis the computation complexity of SAT and HRT in Appendix C.
>
> ### 2. Pareto frontier ###
>
> We list the BLEU score and relative speedup on the WMT En-De task. All speed is measured on a Titian X GPU:
>
> | Model | BLEU | B=1 | B=8 | B=16 | B=32 |
> | :----: | :----: | :----: | :----: | :----: | :----: |
> | AT | 27.45 | x1 | x1 | x1 | x1 |
> | MP4 | 25.94 | x4.58 | x2.91 | x1.51 | x0.81 |
> | MP10 | 27.03 | x1.85 | x1.01 | x0.55 | x0.31 |
> | HRT1-1 | 27.85 | x1.81 | x1.77 | x1.64 | x1.67 |
> | HRT5-1 | 27.98 | x1.59 | x1.59 | x1.52 | x1.53 |
>
> We can see that HRT1-1 is the Pareto frontier compared with MP10 no matter how the decoding batch changes. For MP4,  HRT-1-1 can achieve a faster decoding speed and high BLEU scores than MP-4 when the decoding batch size is larger than 16.  CPU has a similar trend.
>
> ### 3. Speedup less than Levenshetain Transformer & JM-NAT
>
> Since both Levenshetain Transformer and JM-NAT use iterative decoding, both of them have the same problem as demonstrated in our work. However, the reported numbers in their works are measured at batch size 1, which is prone to look like "very fast". We think that directly comparing these numbers may be unfair.
>
> ### 4. Why does HRT1-1 have a different trend from MP1?
>
> Very good point. The main reason is that HRT1-1 uses beam size 1, while MP1 uses beam size 5.
> Please note that because Skip-AT provides a good partial hypothesis, Skip-MP does not need to consider more candidates. In addition, the initial speedup ratio of MP1 is very large (e.g. 8x faster than AT when the batch is 1). In contrast, HRT1-1's speedup at batch 1 is only about 2x. Therefore, it seems that MP1 drops rapidly. Actually,  the speedup ratio of these two models is very close when the batch is 32 (3.2x vs. 4.8x).
>
> ### 5. The advantage of HRT over other acceleration methods
>
> We completely agree with you. We have added more related works in the updated version. Take the simplified decoder as an example, this method still relies on autoregression. In principle, HRT replaces partial autoregressive generation with one-pass non-autoregressive translation. Therefore, HRT is complementary to such pure autoregressive translation models.
>
> ### 6. It’s hard to conclude that the balanced distribution is necessary.
>
> We totally agree with you and have incorporated this suggestion throughout our paper. Please refer to the answer to #R1 for details.
>
> ### 7. Positional encoding in HRT
>
> Yes, we consider the interval k for training Skip-AT of HRT. Taking the Skip-AT's input in Table 1 as an example, the position of ($<s2>, y_2, y_4$) is (0, 2, 4), instead of (0, 1, 2).
>
> ### 8. Apply optimization methods to MP
>
> Yes, some proposed optimization techniques are common. We did this experiment on WMT's En-De task. Due to limited computing resources, the experiment is still in progress, and we will report the results once it is completed.
>
> ### 9. How many runs when measuring the speed?
>
> All speeds are measured by running three times and reporting the average value.
>
> ### 10. wrong numbers in Table 3
>
> Sorry for your confusion. After checking, we found that the numbers for En->Ro translation in Table 3 are reported wrongly in the first submission. We revised the numbers in the updated version. Please note that this change has no effect on the main conclusion, that is, mixed distillation works better than using raw data or distillation data alone in most cases (except En->Ro $C_d$=1).
>
> ### 11. Citation format
>
> We fixed the format problem of the reference in the updated version.
>
> ### 12. Experiment on distant language pairs
>
> We conducted a new experiment on the Chinese-English (Zh-En) translation task. We use the same datasets as [1]: the training data is selected from LDC Corpus and contains 1.8M sentence pairs; NIPS MT06 as the validation set and MT04, MT05, and MT08 as test sets. We train all models for 50k steps and other training setting is the same as WMT En-De. The experimental results are the following:
>
> | System | MT04 | MT05 | MT08 |
> | :---- : | :----: | :-----: | :----: |
> | AT | 43.86 | 52.91 | 33.94 |
> | MP-1 | 23.56 | 31.26 | 17.77 |
> | MP-4 | 40.16 | 50.14 | 31.23 |
> | MP-10 | 42.47 | 52.16 | 33.09 |
> | HRT-1-1 | 43.96 | 53.16 | 33.99 |
> | HRT-5-1 | 44.28 | 53.44 | 34.63 |
> | HRT-5-5 | 44.31 | 53.77 | 34.74 |
>
> We can see that HRT is superior to the original AT model and MP model again, which indicates that HRT is effective in both close and distant language pairs.
>
> =========
> [1] Wang et al. Multi-Layer Representation Fusion for Neural Machine Translation. COLING 2018

---

> > ### Author Response · Authors · 2020-11-23
> > **Supplement to Question 8: Apply optimization methods to MP**
> >
> > We conducted experiments on the WMT En-De task to verify whether these optimization methods are complementary to MP. Please note that the joint training Skip-AT and Skip-MP by curriculum learning is not compatible with MP.  Therefore, we only tested the following methods: including fine-tuning from the pre-trained autoregressive model (+FT),  relative position representation (+RPR), and mixed distillation (+MD).
> >
> > The experimental results are listed here:
> >
> > | Model | Training Step | BLEU |
> > | :----: | :----: | :----: |
> > | MP (official) | 300k | 27.03 |
> > | HRT-5-1 | 100k | 27.98 |
> > | MP + FT + RPR + MD | 100k | 27.02 |
> > | MP + FT + RPR + MD | 160k | 27.32 |
> > | MP + FT + RPR + MD | 300k | 27.63 |
> >
> > We can see that using FT, RPR and MD improves the convergence of model training. E.g., the BLEU of training MP + FT + RPR + MD 100k steps approaches that of the officially released model, which is trained for 300k steps. When training MP + FT + RPR + MD for more steps (e.g, 160k), we can further obtain +0.3 BLEU point improvement. However,  our HRT model trained for 100k steps still outperforms all of them.

---

> > > ### Comment · AnonReviewer3 · 2020-11-25
> > > **Thanks for the added experiments which show HRT outperforms MP, but what about SAT?**
> > >
> > > Thanks for your detailed response! Most of my concerns are addressed, except for my first major concern.
> > >
> > > * Regarding comparison with SAT
> > >
> > >   Only providing the reported result from Wang et al., (2018) (not self implementation and no solid comparison) is not enough in my mind, due to the following reasons:
> > >   1. I have stated in my previous comment: The first part AT in HRT acts very similar to SAT, so SAT *should* be treated as a baseline.
> > >   2. One key advantage of HRT over MP is its compatibility with varying batch sizes. However, this advantage almost disappears when compared to SAT. Wang et al. (2018) also showed that SAT is compatible with different batch sizes.
> > >   3. HRT achieves a decoding speedup of around 1.7x on GPUs across different batch sizes. In similar setting, SAT achieves a speedup of around 1.5x (Wang et al., 2018). The speed difference looks very marginal. Without thorough comparison, it's hard for me to conclude that HRT outperforms SAT. Thus, I can hardly judge the value of this paper.
> > >   4. Showing the decoding complexity as given in Appendix also hardly convinces me. For example, the decoding complexity of MP often looks very appealing, but in this paper, you discovered many practical problems.
> > >   5. The optimization methods seem very promising as shown on WMT En-De task with MP. So these methods have a large chance to improve SAT as well. Comparing highly optimized HRT to the raw SAT model in Table 2 is unfair.
> > >
> > > I tend to keep my original judgement.

---

### Official Review · AnonReviewer2 · 2020-10-28

**Rating:** 7
**Confidence:** 4

**Review:**

This paper proposes a hybrid-regressive machine translation (HRT) approach—combining autoregressive (AT) and non-autoregressive (NAT) translation paradigms: it first uses an AT model to generate a “gappy” sketch (every other token in a sentence), and then applies a NAT model to fill in the gaps with a single pass. As a result the AT part latency is roughly reduced by half compared to a full AT baseline. The AT and NAT models share a majority part of the parameters and can be trained jointly with a carefully designed curriculum learning procedure. Experiments on several MT benchmarks show that the proposed approach achieves speedup over the full AT baseline with comparable translation quality.

The idea of combining AT and NAT is interesting, and the paper is very clearly written. The experiments and analysis are solid and well-designed. I vote for acceptance.

Pros:
- An interesting idea combining AT and NAT which can inspire future research.
- Experiments and analysis support most of the claims.
- The proposed curriculum training and mixed distillation can be useful in future research.

Cons:
- It is a bit sad that the translation quality drops a lot when k > 2. This limits and maximum speedup that can be achieved.
- Some of the claims on the HRT’s performance compared to the AT baseline seems a bit misleading (see details below).

Details:
- From Table 3 it seems that AT models also benefit from mixed distillation. Based on the numbers here I’m assuming the AT models in Table 2 are trained w/o mixed distillation. Early on the paper claims that HRT can outperform AT in terms of translation quality, but this doesn’t seem to be the case if one compares HRT against AT both trained with mixed distillation. Can the authors clarify? If this is true, please tone down the claims on this point in the abstract, intro, and Section 5.
- Can the authors comment on the training cost of HRT in comparison to AT in terms of, e.g., number of epochs, wall-clock time?
- Does the AT baselines in Table 2 use the same relative positional encodings as HRT? If not, can the authors comment on how it may affect their performance?

---

> ### Author Response · Authors · 2020-11-16
> **Response to Reviewer #2**
>
>
> Thanks for your insightful reviews and we appreciate your valuable suggestions! We address your concerns and questions as follows:
>
> ### 1. Translation quality drops a lot when k > 2 ###
>
> Although our synthetic experiments show that a well-trained Mask-Predict model can recover most of the translation performance even if masking 60%-70% (about k=3) tokens of the standard AT hypothesis, it is a great challenge for our Skip-AT to predict the next k-th (k>2) token directly. We found that similar results are reported in previous literatures [1][2]. We think that skip-generation may require more fancy model architecture and training methods, which is left for our future work.
>
> ### 2. Tone down the claims ###
>
> We thank the reviewer for pointing out this issue. We want to clarify that a single HRT model can support both hybrid-regressive decoding ($C_d$=k) and autoregressive decoding ($C_d$=1) at inference. We distinguish the vanilla autoregressive model (V-AT) from HRT with autoregressive decoding (H-AT). Table 3 only shows the effects of different data strategies on HRT, not involving V-AT. As far as HRT itself is concerned, it is true that autoregressive decoding works better than hybrid-regressive decoding. Our previous claim is based on the comparison between V-AT and hybrid-decoding of HRT. Sorry for the confusion, we have updated the related statement in the revised version.
>
> ### 3. Training cost ###
>
> We train HRT for the same epochs as AT. For example, on WMT En->De, we trained both AT and HRT for 100k steps (about 21 epochs). However, it should be noted that we fine-tune HRT on a well-trained AT model for faster convergence, instead of training from scratch.
> Since HRT needs to jointly train Skip-AT and Skip-MP (please see Algorithm 1 in Appendix B), the wall-clock time is about 2 times longer than AT in the same training epochs. As for MP, we use the officially released models of training 300k steps from scratch.
>
> ### 4. Relative positional encoding ###
>
> Very good point. AT baselines in Table 2 are the vanilla Transformer without using relative positional encoding. In the originally submitted version, we have stated in the footnote that "For autoregressive baselines, adding RPR in the Transformer decoder did not bring obvious improvement over the vanilla Transformer. For example, on WMT’14 En→De, Transformer=27.45 and Transformer+RPR=27.34". Primitive experiments show that relative positional encoding has no obvious positive effect in the pure autoregressive translation model.
>
> [1] Wang et al, Semi-Autoregressive Neural Machine Translation. EMNLP 2018
>
> [2] Ran et al, Learning to Recover from Multi-Modality Errors for Non-Autoregressive Neural Machine Translation. ACL 2020

---

> > ### Comment · AnonReviewer2 · 2020-11-17
> > **Thanks for the response!**
> >
> > I have read other reviews as well as the authors' response. I'd like to keep my initial recommendation.
> > I would appreciate it if the revision can be more honest about training cost, and what a small $k$ implies in terms of the limit of speedup that can be achieved.

---

### Official Review · AnonReviewer1 · 2020-10-29
**Interesting findings**

**Rating:** 6
**Confidence:** 5

**Review:**

This paper's main topic is the actual usability of the current status of non-auto-regressive translation (NAT) models.

Although previous papers have reported that the NAT models can achieve the same performance level with auto-regressive translation models while the decoding speed is much faster, like two to five times, this paper points out that it deeply relies on the batch size and computation environment.
This is a proper investigation for the community since some researchers might believe that NAT is always faster than standard auto-regressive models and became an excellent alternative to them.

The ideas of inducing skip-AT and skip-MT are really unique and somewhat innovative (since, I guess, no other researchers hardly think to employ such skip-decoding architecture).
Basically, this paper has several new findings that should be shared in the community for developing better technologies.




The following are the questions/concerns of this paper.

1,

"IR-NAT heavily relies on small batch size and GPU, which is rarely mentioned in prior studies."
I think this is an excellent investigation. However, this paper does not tell readers why this observation happens.
Please explain why the current NAT models are not suitable to work on CPUs and large batches.


2,

The intention of the statement, "which indicates that the balanced distribution of deterministic tokens is necessary" is unclear. Please elaborate on what the authors try to tell by this statement.


3,

The proposed method consists of many new components.
The authors provided the results of an ablation study in Table 5.
This is a really nice analysis.
However, the performance differences in −FT, −RPR, and −MixDistill are somewhat marginal.
Actually, we can easily observe such 0.3-0.4 BLEU score difference by just changing random seeds for Transformer models.
Are there any statistically significant differences among them? Or any reasonable evidence that supports the difference?


4,

This is just a comment, and appreciate having the author's words.

There is an opinion that fully tuned implementation for standard auto-regressive models outperforms both decoding speed and accuracy. See the following presentation slide on WNGT-2020 (such as P33):
https://kheafield.com/papers/edinburgh/wngt20overview_slides.pdf

What would happen for the proposed method if we compared them on such a highly-tuned implementation?

---

> ### Author Response · Authors · 2020-11-16
> **Response to Reviewer #1**
>
>
> Thanks for your insightful reviews and we appreciate your valuable suggestions!
>
> We address your concerns and questions as follows:
>
> ### 1. Why current NAT models are not suitable to work on CPUs and large batches? ###
>
> We thank the reviewer for pointing out this issue. In short, we suspect that for current devices, when the computation cost is high enough, the actual acceleration of parallel computing from the NAT model will slow down.
>
> Specifically, for the NAT model, we assume that the computation cost of each iteration is proportional to the input tensor $(BH \times BM, L, H)$, where BH is the batch size, BM is the beam size, L is the predicted target length, and H is the network dimension. In this way, the total cost of N (generally, N<L) iterations is $C_{nat} \propto N \times O(BH \times BM \times L \times H)$.  For convenience, we omit BM and H in $C_{nat}$ and we simplify $C_{nat}$ to $N \times O(BH \times L)$. Likely,  the computational cost for AT model is about $C_{at}=L \times O(BH \times 1)$ (only the decoder self-attention needs to consider the previous i tokens. We omitted it for clarification.). Then, we can denote the speedup ratio $r=C_{at}/C_{nat} = (L/N) \times (O(BH \times 1)/O(BH \times L))$. Therefore, fewer iterations and faster parallel computation are the keys to IR-NAT.
>
> Direct evidence supporting our view is that when decoding a constant test set (WMT En->De, newstest14), the time spent by AT decreases with the increase of batch size, while MP(I=10) can not benefit from larger batch size (We test on Titian X GPU, and report the average value of 3 runs. BM=5. Note that CPU has the similar results.):
>
> | model | BH=1 | BH=8 | BH=16 | BH=32 |
> | :----: | :----: | :----: | :----: | :----: |
> | AT | 962s | 151s | 78s |  43s |
> | MP (I=10) | 464s | 125s | 119s |129s |
>
> We can see that when BH increases from 1 to 32, AT's latency reduces by 962/43 (about 22) times, while MP (I=10) only reduces by 464/129 (about 4) times. It means that $O(BH \times 1)/O(BH \times L)$ becomes lower as BH increases. Until $O(BH \times 1)/O(BH \times L) < N/L$, NAT will start to be slower than AT.
>
> ### 2. The balanced distribution of deterministic tokens is necessary? ###
>
> Very good point. In Figure 2, we show that "Chunk" has higher BLEU scores than "Random", even if masking more tokens. For example, on the WMT En-De task, when the chunk size is 4 (the masking rate is 75%), the BLEU score of "Chunk" is 27.03, which is +0.3 BLEU higher than that of "Random" with the masking rate of 70%.  We attribute this result to the fact that "Chunk" makes the unmasked tokens more evenly in the sequence. Assuming that 50% of tokens are masked, "Chunk" can guarantee that there are two unmasked tokens (called deterministic tokens) around each <mask> (except for the first and the last <mask>). However, "Random" cannot provide such a guarantee. Considering the extreme case, when all <mask> happen to concentrate on the left/right side of the sequence, "Random" will degrade into "Head"/"Tail". To this end, we stated that "the balanced distribution of deterministic tokens is necessary".
>
> However, as pointed out by Reviewer 3, our statement may be too strong because the gap between "Chunk" and "Random" is not significant. We follow this suggestion and rewrite related sentences. Please see the updated version for details.
>
> ### 3. Ablation study ###
>
> Although it is not obvious in terms of the BLEU score to exclude one component except -MixDistill, we have to emphasize that these components are orthogonal. To verify our statement, we add a new experiment in ablation study, which removes all components, including -FT, -RPR, -MixDistill, and -CL. This model has a BLEU score of 32.57 on the WMT En-Ro task, which falls behind the complete HRT model 1.96 BLEU points, which is very close to the accumulated dropped BLEU (2.13) of applying {-FT, -RPR, -MixDistill, -CL} independently.
>
> ### 4. Apply the proposed method on a highly-tuned AT system ###
>
>  Although we didn't know the listed WNGT-2020 slide before, we found that this slide is very good evidence, which supports our observation that NAT/IR-NAT can hardly have lower latency than AT when using large batch size.
> As shown in P30, in the original NAT study (Gu et al, ICLR 18), the latency of AT was amplified to some extent because they only consider the case of batch size 1. The slide also tells us that when using larger batch sizes, AT system can easily obtain lower latency measured by the average time of translating a single sentence.
> In contrast, our work further shows that the speed degradation of NAT is more severe when using iterative decoding. And we proposed HRT to alleviate this problem by combining the best of both worlds of AT and NAT. We can see that HRT and AT are complementary, so we believe HRT can also accelerate the highly tuned AT system.

---

> > ### Comment · AnonReviewer1 · 2020-11-23
> > **Official Blind Review #1**
> >
> > Thank you for providing detailed answers to my questions.
> > The response answered most of my concerns.
> > I basically feel positive about this paper.

---

### Comment · Area_Chair1 · 2020-11-21
**The discussion stage is open!**

Dear Reviewers:

Thanks for your insightful reviews! Now the discussion stage is open and the authors have posted their responses. We will appreciate that the following things-to-do can be done by Tues, Nov 24.

1 Acknowledge explicitly that you have read the responses.

2 Modify your review if necessary.

3 Communicate with the authors/reviewers/AC by adding/responding to the comments if necessary.

Thanks a lot!

---

### Decision · Program_Chairs · 2021-01-07
**Final Decision**

**Decision:**

Reject

**Comment:**

This paper proposes a new method to combine non-autoregressive (NAT) and autoregressive (AT) NMT. Compared with the original iterative refinement for non-autoregressive NMT, their method first generates a translation candidate using AT and then fill in the gap using NAT.

All of the reviewers think the idea is interesting and this research topic is not well-studied. However, the empirical part did not convince all the reviewers. The revised version and response is good; however, it still does not solve some major concerns of reviewers.